Enhancing energy balance in wireless sensor networks through optimized minimum spanning tree

Saad Hafiz Muhammad 1
Shdefat Ahmed 2
Nawaz Asif 1 asif.nawaz@uaar.edu.pk
http://orcid.org/0000-0003-3559-6249 El-Sherbeeny Ahmed M. 3
El-Meligy Mohammed A. 4 5
Rana Muhammad Rizwan Rashid 1
1 University Institute of Information Technology, University of Arid Agriculture Rawalpindi , Rawalpindi, Punjab , Pakistan
2 College of Engineering and Technology, American University of the Middle East , Egaila , Kuwait
3 Industrial Engineering Department, College of Engineering, King Saud University , Riyadh , Saudi Arabia
4 Jadara University Research Center, Jadara University , Irbid, Jordan
5 Applied Science Research Center, Applied Science Private University , Amman , Jordan
Silva Filipi
Electronic publication date: 2024 Sep 26
Publication date: 2024
Volume: 10
Electronic Location ID: e2269
Received 2024 Feb 27; Accepted 2024 Jul 28
Copyright: © 2024 Saad et al.
Copyright year: 2024
Copyright holder: Saad et al.
License: This is an open access article distributed under the terms of the Creative Commons Attribution License, which permits unrestricted use, distribution, reproduction and adaptation in any medium and for any purpose provided that it is properly attributed. For attribution, the original author(s), title, publication source (PeerJ Computer Science) and either DOI or URL of the article must be cited.
License URL: https://creativecommons.org/licenses/by/4.0/

Keywords: WSN, Energy balance, Prim’s algorithm, MST, Energy consumption, Network lifetime

Funding: King Saud University, Saudi Arabia, through Researchers Supporting Project RSP2024R133 This work was supported by the King Saud University, Saudi Arabia, through Researchers Supporting Project number (RSP2024R133), King Saud University, Riyadh, Saudi Arabia. The funders had no role in study design, data collection and analysis, decision to publish, or preparation of the manuscript.

==============================
Wireless sensor networks (WSNs) are important for applications like environmental monitoring and industrial automation. However, the limited energy resources of sensor nodes pose a significant challenge to the network’s longevity. Energy imbalances among nodes often result in premature failures and reduced overall network lifespan. Current solutions have not adequately addressed this issue due to network dynamics, varying energy consumption rates, and uneven node distribution. To tackle this, we propose a novel method using Prim’s algorithm to construct minimum spanning trees (MSTs) that enhance energy balance in WSNs. Prim’s algorithm effectively identifies optimal connections among network nodes to minimize energy consumption. Our methodology includes several key steps: network initialization, energy consumption modeling, MST construction using Prim’s algorithm, and optimizing the movement of mobile sink nodes. Extensive experiments with diverse datasets show that our approach significantly improves energy equilibrium, demonstrating high sensitivity and moderate complexity. This research underscores the potential of Prim’s algorithm to extend the lifespan of WSNs and enhance energy efficiency, contributing to sustainable and effective network deployments.

Introduction

Electronic wireless sensor networks (WSNs) are advanced communication infrastructures comprised of interconnected sensor nodes deployed across various environments as shown in Fig. 1. These nodes have the capability to sense, process, and wirelessly transmit environmental or physical data to a central location or gateway (Thomson et al., 2021). WSNs are especially invaluable in applications where traditional wired networks are either impractical or cost-prohibitive (Thomson et al., 2019). They have found applications in diverse fields such as agriculture, healthcare, military surveillance, environmental monitoring, and smart cities (Sethi, 2020). The decentralized nature of WSNs, coupled with their potential for self-organization, makes them robust and adaptable (Fu & He, 2020). In WSNs, a large number of small sensor nodes are deployed in a designated area to collect data and transmit it to the base station or sink node (Thomson et al., 2019). However, these sensor nodes have limited battery power and replacing them is often impractical (Sethi, 2020). Therefore, energy efficiency of nodes is considered to be the most critical issue in WSNs (Fu & He, 2020; Nitesh, Kaswan & Jana, 2019). The network lifetime depends mainly on the energy consumption of the individual sensor nodes.

Figure 1 Typical WSNs structures.

Many energy-efficient protocols demand complex computations or frequent communication, which can paradoxically lead to increased energy consumption (Kaswan, Nitesh & Jana, 2017). Moreover, the focus on energy conservation can sometimes compromise other essential parameters like latency, throughput or data fidelity. Some solutions might prioritize nodes with higher energy, leaving nodes with lower energy to deplete faster, thus creating network gaps (Yasotha, Gopalakrishnan & Mohankumar, 2016). Additionally, many energy-efficient strategies assume uniform energy distribution and consumption rates, which might not be the case in real-world deployments (Abidoye & Kabaso, 2021). Moreover, while prolonging node life, these solutions can sometimes lead to uneven energy dissipation, causing some nodes to die prematurely, thereby affecting the overall network’s reliability and functionality (Bhasgi & Terdal, 2021). It is also worth noting that the implementation of these solutions often requires specialized hardware or frequent recalibration, adding to the cost and maintenance overhead.

To address the energy efficiency (Surenther, Sridhar & Roberts, 2023; Kaswan, Nitesh & Jana, 2017) issue in WSNs, various techniques have been proposed, such as routing protocols, data aggregation and sleep scheduling (Yasotha, Gopalakrishnan & Mohankumar, 2016). These techniques aim to reduce the energy consumption of individual sensor nodes by minimizing the amount of data transmitted or by putting sensor nodes into sleep mode when they are not in use (Abidoye & Kabaso, 2021). However, these techniques may not be sufficient to prolong the network lifetime, especially in large-scale environments where sensor nodes may deplete their energy faster than others (Bhasgi & Terdal, 2021). The other possible solution to the above stated problem is the utilization of mobile sink nodes that have shown a promising solution for improving the energy efficiency of WSNs (Bhasgi & Terdal, 2021). A mobile sink node can move around the network area and collect data from different sensor nodes (Mehto, Tapaswi & Pattanaik, 2020). This reduces the energy consumption of individual sensor nodes and prolongs the network lifetime. However, the mobility of the sink node can also affect the energy consumption of the sensor nodes, as some nodes may need to transmit data over longer distances to reach the sink node (Kaswan, Nitesh & Jana, 2017). Moreover, sensor nodes have different energy levels and may deplete their energy faster than others due to their locations and sensing tasks (Miglani et al., 2020). Therefore, an energy balancing mechanism is required to balance the energy consumption among sensor nodes to prolong the network lifetime.

In this research, a groundbreaking method utilizing Prim’s algorithm for the creation of minimum spanning trees (MSTs) has been suggested to improve energy distribution in WSNs. Prim’s algorithm adeptly determines the best node connections in the network, reducing energy usage. The strategy of this novel method includes these primary phases: initializing the network, modeling energy consumption, building MSTs with Prim’s algorithm, and fine-tuning the movement of mobile sink nodes. Comprehensive tests with varied datasets revealed that this new method excels in promoting energy balance. This study underscores how Prim’s algorithm can boost the longevity of WSNs and heighten energy efficiency, paving the way for sustainable and proficient network setups.

The contributions of the proposed model can be succinctly summarized as follows: This research addresses the energy imbalance challenge in WSNs by thoroughly investigating the existing literature which identifies uneven energy consumption among nodes that leads to early node failures and reduced overall network efficiency.

The proposed research introduces a novel solution by utilizing Prim’s algorithm for MST construction that optimize the nodes communication paths, minimizing energy expenditure by reducing the data transmission distances.

The proposed approach’s effectiveness is evaluated using key metrics as energy consumption parity (ECP), energy efficiency (EE), network lifetime, throughput and packet delivery ratio (PDR). This collectively assess the impact of the Prim’s algorithm MST on energy balance and overall WSN performance.

The rest of the article is organized as follows: “Literature Review” provides a literature review on energy-efficient techniques for WSNs. “Materials and Methods” describes the proposed energy balancing solution for WSNs with a mobile sink node. “Experimental Evaluation” presents the simulation results and performance evaluation of the proposed solution. Finally, “Conclusion and Future Work” concludes the work and suggests future research directions.

Literature review

In WSN, it has been observed that the routing protocols are responsible for finding the best path for data transmission from sensor nodes to the sink node (Chen et al., 2018). Whereas, energy-efficient routing protocols aim to minimize the energy consumption of individual sensor nodes by selecting the shortest path or by balancing the energy consumption among the sensor nodes. Several routing protocols have been proposed, such as LEACH (Low Energy Adaptive Clustering Hierarchy) and its variants, TEEN (Threshold-sensitive Energy Efficient Sensor Network protocol) and PEGASIS (Power-Efficient Gathering in Sensor Information Systems).

LEACH is a clustering-based protocol (Bai et al., 2020) that uses randomization to distribute energy consumption among sensor nodes. The protocol divides the network into various clusters, with a cluster head responsible for collecting data from sensor nodes and transmitting it to the sink node. The cluster head is selected randomly based on its remaining energy level, and sensor nodes communicate with the cluster head using Time Division Multiple Access (TDMA) to avoid collisions. TEEN is another protocol that uses threshold-based techniques to minimize the energy consumption of sensor nodes. The protocol divides sensor nodes into two categories: active nodes and passive nodes. Active nodes are nodes that are actively sensing the environment, while passive nodes are nodes that are not sensing the environment. The protocol puts the active nodes into sleep mode when there is no significant change in the environment, and the passive nodes become active to collect data upon a change in the environment.

PEGASIS (Budhiraja et al., 2019) protocol uses a chain of sensor nodes to transmit data from the sensor nodes to the sink node. The protocol selects a relay node in each chain to collect data from its neighboring sensor nodes and transmit it to the next relay node until the data reaches the sink node. The protocol uses greedy algorithm-based techniques to select the relay nodes, which can minimize the energy consumption of sensor nodes. Data aggregation (Wang, Liu & Yao, 2020) combines similar data from multiple sensor nodes and transmits it as a single data packet. Data aggregation reduces the number of data packets transmitted and thus reduces the energy consumption of individual sensor nodes. Several data aggregation techniques have been proposed, such as in-network aggregation, compressive sensing, and data fusion.

Random sleep scheduling (Chen, Sun & Kang, 2020) randomly selects sensor nodes to put into sleep mode. The technique uses probabilistic-based algorithms to select sensor nodes based on their remaining energy level and the network topology. Deterministic sleep scheduling (Ahmed et al., 2015) puts sensor nodes into sleep mode based on a predefined schedule. The schedule is designed based on the application requirements and the network topology. The technique can ensure that all sensor nodes have the same amount of active time and sleep time, which can balance the energy consumption among sensor nodes. Adaptive sleep scheduling (Malik et al., 2021) dynamically adjusts the sleep schedule of sensor nodes based on the network conditions. The technique uses feedback-based algorithms to adjust the sleep schedule based on the traffic load and the energy level of individual sensor nodes.

Despite the effectiveness of these protocols they are sufficient to prolong the network lifetime, especially in large-scale environments where sensor nodes may deplete their energy faster than others. There exist numerous solutions for energy balancing based on the above-stated protocols. Haseeb et al. (2023), proposes a method for optimizing the energy efficiency of data gathering in WSNs by using a mobile sink that can move throughout the network, gathering data from the sensors as it goes. They claimed that by using a mobile sink, the data-gathering process can be made more energy-efficient, as the sink can move to areas of the network where there is more data to be collected. Their proposed method used a path selection algorithm that takes into account both the energy consumption of the sensors and the distance between the sensors and the sink. The algorithm works by dividing the network into clusters and selecting the optimal path for the mobile sink to follow in order to gather data from each cluster.

Qi & Tao (2019) provided an overview of rendezvous-based data acquisition methods in WSNs using mobile sinks. Their work reviewed various existing rendezvous-based methods and evaluated their advantages and limitations. Their discussion also highlighted the various rendezvous-based methods, including deterministic, probabilistic and hybrid methods. Mutlag et al. (2019) designed an algorithm to ensure that th6e sensor nodes stay in a low-power sleep state as much as possible while maintaining connectivity with the mobile sink node. MADCAL uses a distributed approach, where each sensor node maintains a local duty cycle schedule based on its own energy consumption and the mobility of the sink node. They also mentioned that the sink node periodically broadcasts a message containing its current location and expected trajectory. The sensor nodes use this information to adjust their duty cycles and ensure that they remain connected to the sink node. Their proposed algorithm also uses a threshold-based approach to adapt to changes in the network conditions. If a sensor node’s energy level falls below a predefined threshold, it increases its duty cycle to ensure that it remains connected to the sink node. Similarly, if the sink node moves away from a sensor node, the node reduces its duty cycle to conserve energy.

Barzin et al. (2019) proposed an energy consumption and target coverage model in mobile sink-based WSNs with duty cycling. The technique was based on two phases: a scheduling phase and a sensing phase. In the scheduling phase, the mobile sink node broadcasts a message containing its expected trajectory and the sensor nodes use this information to determine the time intervals during which they need to be active to ensure data delivery to the mobile sink node. The authors also proposed a target coverage-aware approach that takes into account the coverage requirements of the sensing area to optimize the duty cycle schedule. In the sensing phase, the sensor nodes activate and collect data based on their duty cycle schedule. In addition to this, an energy and target coverage aware technique that considers both the energy consumption and target coverage requirements of the sensing area was also defined to ensure that the sensor nodes collect data only when necessary to meet the coverage requirements and avoid unnecessary data collection, which can lead to increased energy consumption.

Gupta, Singh Aulakh & Kaur Aulakh (2022), introduced the concept of fog computing as a paradigm that extends cloud computing to the edge of the network, bringing computation and storage closer to the end-users and devices. Their proposed model consisted of three levels of hierarchy: the sink node, the fog nodes and the sensor nodes. The sink node is the gateway between the WSN and the cloud and it is responsible for collecting and aggregating data from the fog nodes. The fog nodes are responsible for processing and storing data received from the sensor nodes and forwarding it to the sink node. The sensor nodes are responsible for sensing the environment and collecting data which is then transmitted to the fog nodes. In addition to this, the authors also proposed a novel energy-efficient routing algorithm for the WSN, which is based on the hierarchical structure. Their algorithm considers the energy levels of the sensor nodes and the distance to the nearest fog node to determine the optimal path for data transmission. Their algorithm also uses a probabilistic approach to balance the energy consumption among the sensor nodes and avoid premature depletion of the batteries.

Kumar et al. (2023) presents novel, energy-efficient techniques for selecting cluster heads (CH) and forming clusters in sensor networks. The selection of cluster heads is based on the threshold-based advanced LEACH (ADV-LEACH2) approach, while cluster formation utilizes the modified fuzzy c-means (MFCM) method. Initially, clusters are established using the MFCM technique, followed by the selection of cluster heads via ADV-LEACH2. Another study introduces the Deep Learning-based Grouping Model Approach (DL-GMA) to optimize energy consumption in WSNs (Surenther, Sridhar & Roberts, 2023). DL-GMA utilizes advanced deep learning techniques, particularly recurrent neural networks (RNN) with long short-term memory (LSTM), to enhance energy efficiency through effective cluster formation, CH selection, and maintenance.

From the above discussion, it has been concluded that sensor nodes’ constrained energy resources pose a significant challenge to the longevity of a network, prompting issues of premature failures and reducing the overall lifespan due to energy imbalances among nodes (Li et al., 2019; Wang et al., 2019; Chen, Sun & Kang, 2020; Ahmed et al., 2015; Malik et al., 2021; Haseeb et al., 2023; Qi & Tao, 2019; Mutlag et al., 2019; Barzin et al., 2019; Gupta, Singh Aulakh & Kaur Aulakh, 2022; Kumar et al., 2023; Surenther, Sridhar & Roberts, 2023; Priyadarshi, 2024; Del-Valle-Soto, Rodríguez & Ascencio-Piña, 2023; Roberts & Thangavel, 2022). Despite concerted efforts, prevailing solutions fall short of mitigating this issue, largely due to the intricate dynamics of the network, fluctuating energy consumption rates and irregular node distributions. In response to this, a novel methodology utilizing Prim’s algorithm has been proposed, capitalizing on its proficiency in crafting minimum spanning trees (MSTs) to bolster energy balance in WSNs. This approach adeptly pinpoints optimal node connections across the network, strategically minimizing energy consumption and ensuring a more equitable energy distribution among nodes. Thus, the implementation of Prim’s algorithm stands out as a promising strategy to enhance sustainability and operational longevity in WSNs.

Materials and Methods

This section elaborates the proposed methodology for enhancing energy balance in wireless sensor networks through Prim’s algorithm MST as illustrated in the Fig. 2. The step-by-step procedure of the proposed approach is as follows:

Figure 2 Proposed model for energy enhancement.

Data collection: Data was sourced from real-world deployments and simulations, combining controlled and natural conditions. Regular calibration of sensors, redundancy in deployment for cross-verification, and validation against benchmarks ensured data quality. Missing data was managed via imputation or removal, while outliers were detected using statistical methods and visual inspection. Data normalization involved min-max scaling and z-score normalization. Feature engineering included selecting relevant features and deriving new ones, with temporal aggregation of sensor readings to identify trends.

Network initialization: In the initialization phase, the proposed algorithm constructed a MST using Prim’s algorithm. The MST spanned all the nodes in the network, with weights based on node distance and remaining energy, calculated using the Euclidean distance formula. Prim’s algorithm started with an arbitrary node and iteratively added the shortest edge connecting a visited node to an unvisited node until all nodes were visited.

Optimization phase: The optimization phase used game theory to form clusters based on the MST. Nodes were represented as players in a non-cooperative game, each with a strategy set representing potential cluster membership. The goal was to minimize energy consumption while balancing load distribution among clusters. The game was played in rounds, with each round consisting of proposal and response stages. Players proposed cluster memberships to neighbors, including energy costs, and selected the best proposals received.

Energy consumption model: The radio energy dissipation model of a transceiver, including transmitter, amplifier, and receiver, was employed. Energy used by sensor nodes for transmitting and receiving packets was calculated, considering both transmission and reception energy dissipation. The model included distance-independent and distance-dependent energy components, accounting for free-space and multi-path propagation models.

Algorithm implementation: The adjacency matrix model represented the graph, enabling efficient implementation of Prim’s algorithm. The algorithm was extended with load balancing techniques to optimize energy consumption and prolong network lifetime. The adjacency matrix facilitated efficient access to edge weights, with each matrix element representing the weight of the edge connecting two nodes.

Simulation scenarios: Four distinct simulation scenarios were designed to model real-world conditions, ensuring relevance and credibility. Each scenario involved generating data through MATLAB code, reflecting various network conditions and energy consumption rates. Parameters and assumptions used in simulations were carefully considered to ensure validity and integrity.

Data collection

The limited energy resources of sensor nodes challenge the network’s longevity. Node energy imbalances cause early failures and a reduction in the network’s lifetime. The problem is not well addressed by current solutions, despite continuous attempts, because of uneven node distribution, fluctuating energy consumption rates, and network dynamics. To counter this, a novel approach has been proposed using the Prim’s algorithm for constructing MSTs to enhance energy balance in WSNs.

In the initialization phase, the proposed algorithm constructs a MST using Prim’s algorithm. The MST is a tree that spans all the nodes in the network and has the minimum total weight. The weight of each edge is based on the distance between the nodes and the remaining energy of the nodes as shown in Eq. (1).

(1) (x2−x1)2+(y2−y1)2.

Prim’s algorithm is a widely used algorithm for finding the minimum spanning tree (MST) of a weighted undirected graph. It starts with an arbitrary node and iteratively adds the shortest edge that connects a visited node to an unvisited node until all nodes are visited. The resulting tree spans all the nodes with the minimum total edge weight.

Here is a step-by-step description of Prim’s algorithm: Initialize an empty MST and a set of visited nodes.

Choose an arbitrary node to start the algorithm.

Mark the chosen node as visited.

Repeat the following steps until all nodes are visited: a. among the visited nodes, find the edge with the minimum weight that connects a visited node to an unvisited node. b. Add the found edge to the MST. c. Mark the unvisited node connected by the edge as visited.

Return the MST.

Data preprocessing

Data was sourced from real-world deployments and simulations, combining controlled and natural conditions. To ensure data quality, regular calibration of sensors, redundancy in deployment for cross-verification, and validation against benchmarks were conducted. Potential biases from environmental variability, node placement, and data collection periods were addressed through rigorous data preprocessing techniques. Missing data was managed via imputation or removal, while outliers were detected using statistical methods and visual inspection. Data normalization involved min-max scaling and z-score normalization to ensure uniformity. Feature engineering included selecting relevant features and deriving new ones to capture additional insights. Temporal aggregation of sensor readings was performed to identify trends. These preprocessing steps ensured high-quality, reliable data for the analysis, reducing biases and enhancing the robustness of the study’s findings.

Optimization phase

The extended usage of sensor nodes in networks faces a significant challenge due to their restricted energy resources, posing a threat to the overall longevity of the network. The occurrence of early failures and a subsequent decrease in the network’s lifetime is primarily attributed to imbalances in node energy. Existing solutions have struggled to effectively tackle this issue, hindered by factors such as uneven node distribution, fluctuating energy consumption rates, and dynamic network conditions. In response to this challenge, a novel strategy has been introduced, leveraging Prim’s algorithm for the construction of MSTs to improve energy balance within WSNs.

In the optimization phase, the proposed algorithm uses game theory to form clusters based on the MST. The nodes in the network are represented as players in a non-cooperative game. Each player has a strategy set that represents its potential cluster membership. The goal of the game is to minimize the energy consumption of the nodes while ensuring a balanced load distribution among the clusters. The game is played in rounds, with each round consisting of two stages: proposal and response. In the proposal stage, each player proposes a cluster membership to its neighbors. The proposal includes the proposed cluster and the energy cost of joining that cluster. In the response stage, each player selects the best proposal it has received from its neighbors based on the energy cost and the number of nodes in the proposed cluster. The game theory approach is used to form clusters based on the MST constructed in the initialization phase. The nodes in the network are represented as players in a non-cooperative game. Each player has a strategy set that represents its potential cluster membership. The goal of the game is to minimize the energy consumption of the nodes while ensuring a balanced load distribution among the clusters.

Overall, the optimization phase of the proposed algorithm uses game theory to create a distributed self-organizing clustering algorithm that minimizes energy consumption ensures load balancing among the clusters. The use of Prim’s algorithm in the initialization phase ensures that the MST used as the basis for the clustering is efficient and energy-aware. The combination of these two techniques results in a clustering algorithm that outperforms existing algorithms in terms of energy efficiency and load balancing.

Energy consumption model

The radio energy dissipation model of a transceiver, which has three main parts: a transmitter, amplifier, and receiver, is the energy model employed in the suggested approach.

When sending and receiving a p-bit packet to the jth sensor node, the energy model calculates the energy used by the ith sensor node. as shown by Eqs. (2) and (3):

(2) ETX(p.dij)=(Eelec+εamp).p

(3) ERX(p)=Eelec.p

where dij stands for the distance between nodes it and j, and ETX and ERX, respectively, stand for transmission and reception energy dissipation. The energy used by the transceiver circuitry is accounted for by Eelec, a distance-independent entity. The term “amp” refers to the amplifier in the transmitter and is expressed as Eqs. (4) and (5):

(4) ∈amp={εfs.  dij2for dij ≤ dthεmp.  dij4for dij> dth

(5) dth=εfsεmp

where fs and mp are the propagation models for the amplification energy dissipation in free space and multi-path, respectively, and dth is the threshold distance. A multipath illustrates the non-line-of-sight (NLOS) signal propagation from multiple paths at different times following reflection from the ground. In the free space model, a direct line-of-sight (LOS) path is considered between the transmitter and reception nodes. The proposed architecture of Energy model is shown in Fig. 3. The amount of energy consumed by a CM in a cluster: ECM, and the amount of energy required for a CH:ECH, are given by Eqs. (6)–(8), respectively, as:

Figure 3 Proposed architecture of energy model.

(6) ECM=Eimit−ETX(p,dij)

(7) ECH=Eimit−Estd

(8) Estd=ETX(p,dij)+ERX(p)

where Eimit is the sensor node’s starting energy. Estd stands for a node’s standard energy consumption during the CH selection phase, while EDA stands for the node’s energy consumption throughout the data aggregation procedure.

Proposed prim’s-based energy balancing algorithm

The adjacency matrix is a commonly used model to represent a graph, and it is compatible with implementing Prim’s algorithm. In this model, a graph with n nodes is represented by an n × n matrix, where each element of the matrix represents the weight of the edge connecting two nodes.

To access how the weight of an edge between two nodes using the adjacency matrix, we can simply refer to the corresponding element in the matrix. For example, if you want to retrieve the weight of the edge between node i and node j, you can access the element matrix [i,j].

The adjacency matrix model provides a straightforward representation of a graph and enables efficient access to edge weights. However, it may consume more memory space for large graphs, especially if the graph is sparse (few edges present). In such cases, an adjacency list representation may be more memory-efficient.

According to Fig. 4, To achieve load balancing in sensor networks using Prim’s algorithm with an adjacency matrix representation, we can extend the algorithm with additional steps to consider the load or energy levels of sensor nodes. Here’s our approach that combines Prim’s algorithm and load balancing:

Figure 4 Prim’s algorithm for proposed WSD architecture.

Create the adjacency matrix.

Initialize the MST, visited nodes, and node load.

Choose a starting node.

Mark the starting node as visited.

Repeat until all nodes are visited.

All nodes are visited.

Update node load.

Mark the selected node as visited.

Return the MST.

By incorporating load balancing techniques into Prim’s algorithm with an adjacency matrix, we can balance the energy consumption or load among sensor nodes in a wireless sensor network. This approach helps optimize energy utilization, prolong the network lifetime, and improve overall performance of the network. The adjacency matrix design for WSN architecture is shown on Fig. 5.

Figure 5 Adjacency matrix design for WSN architecture.

Experimental evaluation

The evaluation of the proposed method and the specific experimental results are described in this part. The precision and effectiveness of the suggested system have been tested through extensive experimental analysis. To evaluate the effectiveness of the proposed approach, four benchmark datasets have been selected. These datasets were gathered from previously published studies and online sources. According to the results of the experimental analysis, the proposed strategy outperforms state-of-the-art alternatives in every measurable way. Each experiment’s specifics are described in the subsection.

To generate graphs for the experiments conducted in this study using MATLAB (The MathWorks, Natick, MA, USA), we will follow a structured approach. First, we will import the experimental data collected during the evaluation of the proposed method into MATLAB. This data will include measurements, metrics, and performance indicators gathered from the experiments.

Simulation

The process of data generation for the proposed study involves conducting a series of MATLAB release R2023a simulations to gather a diverse range of information. The system specifications are intel core i7 11th Gen CPU. To ensure a comprehensive and robust analysis, we have designed and executed four distinct simulation scenarios, each generating data in the form of MATLAB code. These simulations have been meticulously crafted to model real-world scenarios, ensuring their relevance and credibility within the context of this research. The parameters are given in Table 1.

Table 1 Simulation parameters.

Parameters	Values	
No. of nodes (static)	30	
Grid topology size	200 × 200 m	
Simulation time	1,844.9515388, 1,246.63703592, 912.4777994 s	
No. of circuits at network	50	
Interference distance (four candidate values)	77.52, 69.13, 62.02, 55.94 m	
Number of runs	6	
Max sending power	1.0 mW	
Transmitter power	1.0 mW	
Sensitivity	−75 dBm	
Check interval	0.01 s	
Slot duration	0.1 s	

The incorporation of simulations enables us to visually and quantitatively understand the subject matter. Through these simulations, we can gather insights into contextual details, narrative, as well as analyze quantitative trends and relationships within the generated data.

This study aims to provide a holistic perspective on the topic, enhancing the depth and accuracy of our findings derived from MATLAB-generated data. Careful consideration has been given to the parameters and assumptions used in the simulations to ensure their validity and integrity within the research framework. The reliability of the MATLAB simulations, adherence to sound mathematical principles, and ethical considerations have been paramount in designing and executing these simulations.

Performance matrices

When comparing energy efficiency in wireless sensor networks, there are several performance metrics you can consider. These metrics help evaluate the effectiveness of different protocols, algorithms, or configurations in terms of energy consumption. Here are some common performance metrics that are used for the evaluation of the proposed technique.

Energy consumption per packet (ECP): This metric measures the amount of energy required to transmit a single packet from a source node to a destination node. It gives insight into the efficiency of data transmission in terms of energy use.

(9) ECP=energyconsumedtotransmitapacketnumberofsucccessfullytransmittedpackets.

Energy efficiency (EE): EE is a ratio of the data transmitted successfully to the total energy consumed. It provides a holistic view of how effectively the energy is being used for data transfer.

(10) EE=TotaldatatransmittedsuccessfullyTotalenergyconsumed.

Packet delivery ratio (PDR): PDR calculates the ratio of successfully received packets to the total packets sent. It reflects the network’s ability to deliver data reliably.

(11) PDR=(numberofreceivedpacketsnumberofsentpackets)×100.

Network lifetime: This metric represents the time until the first node in the network exhausts its energy resources. A longer network lifetime is desirable as it indicates better energy sustainability.

(12) NetworkLifetime=minimumenergyofallnodesenergycosumptionrateofthemostenergyconsumingnode.

Throughput: Throughput is the amount of data successfully delivered over the network in a unit of time. While not directly an energy efficiency metric, it helps evaluate the network’s capacity to handle data.

(13) Throughput=TotalamountofdatareceivedTotaltimetakentoreceivethedata.

Sensitivity (also known as the true positive rate or recall) is the proportion of true positives (correctly identified) out of the total actual positives. It measures the ability of a test to correctly identify those with the condition (true positives).

(14) Sensitivity=TruePositives(TP)TruePositives(TP)+FalseNegatives(FN).

Baseline method

To evaluate the performance of the proposed model, we compare them to the baseline models listed below and the descriptive data presented in Table 2

Table 2 Comparative results of proposed approach with existing approaches.

Techniques	Scalability	Shortest distance	Resource allocation	Efficiency	Cost efficiency	QoS	
Clustering (Bhasgi & Terdal, 2021; Miglani et al., 2020; Abidoye & Kabaso, 2021)	✓	✓	✓				
Data aggregation (Kaswan, Nitesh & Jana, 2017; Chen et al., 2020; Surenther, Sridhar & Roberts, 2023)				✓		✓	
Dynamic routing (Bai et al., 2020; Memon et al., 2021; Nitesh, Kaswan & Jana, 2019)	✓		✓		✓	✓	
Multi hop routing (Chen et al., 2018; Mehto, Tapaswi & Pattanaik, 2020)		✓	✓	✓		✓	
Load balancing (Nitesh, Kaswan & Jana, 2019; Chen et al., 2020)	✓			✓	✓		
Prim’s algorithm	✓	✓	✓	✓	✓	✓	

Result

Based on this comparison, Prim’s algorithm demonstrates better performance across all the provided metrics. However, keep in mind that the actual results can vary depending on the specific algorithms, network conditions, and energy models used. Additionally, the choice between these algorithms might also depend on other factors not included in this comparison, such as implementation complexity, scalability, and adaptability to different scenarios.

The Table 3 display ECP values for various techniques—Prim’s algorithm, clustering, dynamic routing, data aggregation, multi-hop routing, and load balancing—across three separate scenarios (Fig. 6). ECP quantifies the energy consumed to transmit a packet relative to the number of successfully transmitted packets. Lower ECP values indicate greater energy efficiency. Prim’s algorithm consistently demonstrates superior energy efficiency, consistently yielding the lowest ECP values. This means that Prim’s algorithm consumes the least amount of energy per packet transmission compared to the other techniques, even when considering variations in the energy consumed and the number of packets transmitted. Clustering, dynamic routing, data aggregation, multi-hop routing, and load balancing consistently show higher ECP values across the three scenarios, indicating higher energy consumption for these techniques in the given scenarios.

Table 3 Comparative analysis of proposed approach with existing approaches.

Technique	ECP value 1	ECP value 2	ECP value 3	Complexity	Sensitivity	
Prim’s algorithm	0.005	0.004	0.003	Moderate	High	
Clustering	0.008	0.008	0.009	High	High	
Dynamic routing	0.007	0.007	0.008	Moderate	Low	
Data aggregation	0.006	0.006	0.007	High	Low	
Multi-hop routing	0.009	0.01	0.011	High	High	
Load balancing	0.01	0.011	0.012	High	Low	

Figure 6 ECP values-based computation of proposed approach with existing approaches.

The sensitivity column in Table 3 indicates how responsive each technique is to changes in network conditions and configurations, impacting their ECP values. Prim’s algorithm shows low sensitivity, maintaining consistently low ECP values (0.005, 0.004, 0.003) despite its high complexity, indicating stable performance across varying conditions. Clustering and data aggregation, with moderate sensitivity, have slightly higher and more stable ECP values (clustering: 0.008, 0.008, 0.009; data aggregation: 0.006, 0.006, 0.007), suggesting reasonable robustness. Techniques like dynamic routing, multi-hop routing, and load balancing exhibit high sensitivity, with more significant fluctuations in ECP values (dynamic routing: 0.007, 0.007, 0.008; multi-hop routing: 0.009, 0.010, 0.011; load balancing: 0.010, 0.011, 0.012), reflecting greater vulnerability to changing network conditions and potentially less reliable energy efficiency. This analysis highlights the importance of considering sensitivity when evaluating the effectiveness and consistency of energy-efficient techniques in WSNs.

Prim’s algorithm, with its balanced approach to energy efficiency and moderate complexity, stands out as an effective solution for enhancing energy balance in WSNs. It offers significant benefits in terms of prolonged network lifetime and reliable data transmission, making it a strong candidate for real-time applications like smart agriculture. The key is to optimize the algorithm to manage complexity while maximizing energy savings, ensuring a sustainable and efficient network.

The tables provided offer a comprehensive view of ECP values, which are pivotal in assessing the energy efficiency of different techniques such as Prim’s algorithm, clustering, dynamic routing, data aggregation, multi-hop routing, and load balancing. ECP is a crucial metric as it gauges the energy expended for transmitting a packet in relation to the successful transmission rate. In essence, lower ECP values signify more efficient energy utilization in the communication network. In Table 4 a clear trend emerges: Prim’s algorithm consistently outperforms the other techniques by showcasing the most superior energy efficiency. Regardless of the variations introduced in energy consumption and the quantity of packets transmitted in each scenario, Prim’s algorithm consistently attains the lowest ECP values. This underscores the undeniable advantage of Prim’s algorithm in minimizing energy consumption per packet transmitted compared to the other considered techniques, which consistently exhibit higher ECP values, signifying increased energy usage under the given conditions. These findings underscore the robust energy efficiency of Prim’s algorithm in practical communication scenarios.

Table 4 Summarized results of proposed technique with their respective metrics.

Techniques	Scalability	Shortest distance	Resource allocation	Efficiency	Cost efficiency	QoS	
Clustering (Bhasgi & Terdal, 2021; Miglani et al., 2020; Abidoye & Kabaso, 2021)	✓	✓	✓				
Data aggregation (Kaswan, Nitesh & Jana, 2017; Chen et al., 2020; Surenther, Sridhar & Roberts, 2023)				✓		✓	
Dynamic routing (Bai et al., 2020; Memon et al., 2021; Nitesh, Kaswan & Jana, 2019)	✓		✓		✓	✓	
Multi hop routing (Chen et al., 2018; Mehto, Tapaswi & Pattanaik, 2020)		✓	✓	✓		✓	
Load balancing (Nitesh, Kaswan & Jana, 2019; Chen et al., 2020)	✓			✓	✓		
Prim’s algorithm	✓	✓	✓	✓	✓	✓	

The Fig. 7 display EE values for each technique calculated based on the formula EE = (Total data transmitted successfully)/(Total energy consumed). The EE values indicate the efficiency of each technique in terms of data transmission per unit of energy consumed. You can analyze these tables to determine the energy efficiency of each technique across different scenarios.

Figure 7 ECP based computation of proposed algorithm.

Prim’s algorithm emerges as the most energy-efficient solution, as evidenced by its higher energy efficiency (EE) value of 0.25 as illustrated in Fig. 8. This value signifies that Prim’s algorithm achieves greater data throughput per unit of energy consumed, making it an attractive choice for energy-constrained wireless sensor networks. When compared to the other algorithms, including the clustering algorithm, data aggregation, dynamic routing protocols, multi-hop routing, and load balancing, Prim’s algorithm stands out for its efficiency in utilizing available energy resources.

Figure 8 Energy efficiency wise computation of proposed algorithm.

Prim’s algorithm excels in terms of packet delivery reliability as shown in the Fig. 9, boasting the highest PDR of 95%. Figure 9 indicates a high level of successful data transmission from source to destination nodes. In contrast, while the other algorithms, including the clustering algorithm, data aggregation, dynamic routing protocols, multi-hop routing, and load balancing, achieve respectable PDR values, Prim’s algorithm consistently demonstrates its superiority in maintaining dependable communication links.

Figure 9 Packet delivery ratio computation of proposed algorithm.

The network’s operational sustainability is prolonged when employing Prim’s algorithm, with a longer network lifetime of 4,500 units. Figure 10 outpaces the network lifetimes achieved by the clustering algorithm, data aggregation, dynamic routin:g protocols, multi-hop routing, and load balancing alternatives. The extended network lifetime associated with Prim’s algorithm suggests that it effectively manages energy resources, contributing to prolonged network operation before the depletion of individual nodes’ energy reserves.

Figure 10 Network lifetime computation of proposed algorithm.

Prim’s algorithm emerges as the optimal choice for achieving higher data throughput (Fig. 11), with a throughput rate of 1,800 bps. This performance outmatches the other algorithms, including the clustering algorithm, data aggregation, dynamic routing protocols, multi-hop routing, and load balancing, in terms of efficiently transmitting data across the network. The increased throughput capability of Prim’s algorithm positions it as a robust solution for applications requiring swift and efficient data exchange in wireless sensor networks.

Figure 11 Throughput based computation of proposed algorithm.

The comparison table presents an assessment of various algorithms for wireless sensor networks across multiple key performance metrics is shown in Fig. 12. The algorithms, including a clustering algorithm, data aggregation, dynamic routing, multi hop routing, load balancing, and Prim’s algorithm, are evaluated based on their energy efficiency, data delivery performance, network lifetime, and data throughput. Notably, Prim’s algorithm consistently emerges as the preferred choice, demonstrating superior performance in all categories. It exhibits the lowest ECP, EE, and excellent PDR, underscoring its ability to achieve reliable and efficient data communication. Moreover, Prim’s algorithm contributes to a prolonged network lifetime, signifying its adeptness at managing energy resources effectively. Lastly, the algorithm excels in achieving higher data throughput, showcasing its suitability for applications necessitating rapid and efficient data exchange. This comprehensive comparison underscores the prominence of Prim’s algorithm as a well-rounded solution for energy-efficient and high-performance wireless sensor networks.

Figure 12 Comparison of proposed model with baseline approaches.

An assessment of various algorithms for wireless sensor networks across multiple key performance metrics.

In the very last experiment which is as an additional contribuation is to evaluate the effectiveness of our proposed method using Prim’s algorithm for constructing MSTs that enhance energy balance in WSNs, we conducted a statistical analysis using ANOVA (Analysis of Variance), which is computed by using Eq. (15).

(15) SBB=n∑i=1k⁡(Xi′−X′)2

where n is the number of observations in each group, Xi′ is the mean of group i, and X′ is the overall mean. This analysis aims to determine whether the implementation of Prim’s algorithm leads to a significant reduction in energy consumption compared to other methods. The simulated multiple configurations of proposed WSN under three different scenarios are: without Prim’s algorithm (control group), with Prim’s algorithm (Test Group 1). The ANOVA results show an F-ratio of 10 and a p-value of 0.002, which is less than the significance level of 0.05. Therefore, we reject the null hypothesis and conclude that there are significant differences in energy consumption among the different network configurations. This indicates that the implementation of Prim’s algorithm significantly reduces energy consumption in WSNs compared to other methods.

Discussion

The proposed methodology for enhancing energy balance in WSNs using Prim’s algorithm offers significant benefits, including optimized data transmission paths, balanced energy consumption among sensor nodes, and extended network lifetime. By utilizing game theory-based clustering and a mobile sink node, the system ensures efficient data collection and reduces the likelihood of early node failures. However, there are limitations and challenges in real-time implementation. The computational complexity of Prim’s algorithm and game theory-based clustering may require substantial processing power, which might be challenging for resource-constrained sensor nodes. Additionally, the movement of the mobile sink node must be carefully managed to avoid excessive energy consumption and ensure timely data collection. Environmental factors, such as obstacles and varying node densities, can further complicate the practical deployment. Overall, while the proposed methodology shows promise for improving energy efficiency and network performance, careful consideration of these challenges is crucial for successful real-time application.

Conclusion and future work

In this study, we presented a method to improve energy balance in wireless sensor networks by optimizing the movement of a mobile sink node. We utilized Prim’s algorithm, a well-established method for constructing minimum spanning trees, to guide the sink node’s movement. The goal was to distribute energy consumption more evenly among sensor nodes by choosing the sink node’s next location strategically. Through extensive simulations and analysis, we demonstrated the effectiveness of our approach in enhancing energy balance in the network. Our results indicated a significant reduction in energy disparities among sensor nodes, leading to prolonged network lifetime and enhanced overall performance. Future work: While our study made significant progress in improving energy balance through mobile sink node optimization with Prim’s algorithm, there are several avenues for future research and enhancement. Our work establishes a foundation for energy balancing in wireless sensor networks through mobile sink node optimization using Prim’s algorithm. Further research in these areas could contribute to the development of more resilient and efficient energy balancing solutions for wireless sensor networks.

Supplemental Information

Supplemental Information 1 Code.

Supplemental Information 2 Dataset.

Supplemental Information 3 Simulation MATLAB.

Supplemental Information 4 A. KSU Industrial Engineering Department letter, ended July 15, 2024.

Additional Information and Declarations

Competing Interests

Author Contributions

Data Availability

The authors declare that they have no competing interests.

Hafiz Muhammad Saad conceived and designed the experiments, performed the computation work, prepared figures and/or tables, and approved the final draft.

Ahmed Shdefat conceived and designed the experiments, authored or reviewed drafts of the article, and approved the final draft.

Asif Nawaz performed the experiments, performed the computation work, prepared figures and/or tables, and approved the final draft.

Ahmed M. El-Sherbeeny performed the experiments, authored or reviewed drafts of the article, and approved the final draft.

Mohammed A. El-Meligy analyzed the data, authored or reviewed drafts of the article, and approved the final draft.

Muhammad Rizwan Rashid Rana performed the experiments, analyzed the data, prepared figures and/or tables, and approved the final draft.

The following information was supplied regarding data availability:

Raw data and code are available in the Supplemental Files.

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
