# Peer review of "Enhancing energy balance in wireless sensor networks through optimized minimum spanning tree"

_PeerJ Computer Science, doi:10.7717/peerj-cs.2269_

## Round 0.1 · original submission · Major Revisions

Please consider the reviewers' suggestions to improve the paper. In particular the specific points raised by reviewer #4.

Reviewer #4 asks for more details about the initialization procedure. The authors should also discuss issues such as scalability and discuss real-world applications.

**Language Note:** PeerJ staff have identified that the English language needs to be improved. When you prepare your next revision, please either (i) have a colleague who is proficient in English and familiar with the subject matter review your manuscript, or (ii) contact a professional editing service to review your manuscript. PeerJ can provide language editing services - you can contact us at [email protected] for pricing (be sure to provide your manuscript number and title). – PeerJ Staff

Reviewer 1 ·

Basic reporting

Formal results should include clear definitions of all terms and theorems, and detailed proofs.

Experimental design

Methods should be described with sufficient information to be reproducible by another investigator.

Validity of the findings

Impact and novelty not assessed. Meaningful replication encouraged where rationale & benefit to literature is clearly stated.

Reviewer 2 ·

Basic reporting

1. In general , sensor mote will capture the data frequently in real time applications. How for proposed system will support?

2. The idea presented in the paper is valuable but still, significant effort is needed to improve the clarity of the presentation of the idea and the result with some real time applications

3. Simple realtime example can be added with proposed methodology for better understanding

4. Motivation behind this work can be added with an illustrative example.

5. Some of the formulae in the proposed work are available in the published papers. It is necessary to provide appropriate citations.

Experimental design

Analyze the complexity of the system with existing methods for better understanding.

Proposed methodology part to be improved interms of mathematical form

Not finding any step by step process of new methodology with examples

Discuss benefits and limitations of the proposed work. Possible challenges when implement this work in the realtime scenarios to be discussed.

Validity of the findings

The methodology together its flowchart is very simple. How it works well.

Formulations and model-statements are to be improved further

Needs more clarity on the algorithms used

The authors should highlight the novelty of the work and concentrate on writing a better methodology

Reviewer 3 ·

Basic reporting

1. Introduction section: explain the context of the study and state the precise objective. An Introduction should contain the following three parts: Background, The Problem & The Proposed Solution.

2. The abstract should succinctly summarize the key findings and contributions of the paper. Consider revising the abstract to accurately reflect the content of the paper.

3. The literature review could be more comprehensive and include recent relevant studies. Discuss how your work contributes to or differs from existing research in the field.

4. Elaborate on the data collection process, addressing potential biases and ensuring data quality. Include more information on data preprocessing techniques and the rationale behind them.

Experimental design

1. Clarify the step-by-step procedure of the experimental setup to ensure reproducibility. Provide more details on the selection criteria for participants, materials, or software tools.

2. Discuss the choice of control variables and their impact on the study outcomes. Address potential confounding factors and outline strategies employed to mitigate them.

3. Discuss the validation methodology used for the proposed model or method. Include information on reliability measures to ensure the consistency of your results.

4.

Validity of the findings

1. Provide a more detailed interpretation of the results in the context of existing literature. Discuss the significance of observed trends and variations, connecting them to the research question.

2. Conduct sensitivity analyses on key parameters to evaluate their impact on the results. Discuss the sensitivity of your model or method to variations in input parameters.

3. Ensure that results are presented clearly and concisely, with appropriate figures and tables. Discuss the significance of your findings and relate them back to the research question. Consider including a more in-depth discussion of limitations and future work.

Additional comments

1. Clearly articulate the technical innovations or contributions of your work. Discuss how your methodology or findings advance the state-of-the-art in the field. Provide additional technical details on novel algorithms, models, or techniques introduced in the paper.

2. The writing style could be refined for better clarity and precision. Proofread the manuscript for grammatical errors and typos. Consider using more concise language to improve the overall readability.

Reviewer 4 ·

Basic reporting

The article is clearly written and structured, providing a comprehensive overview of the problem of energy imbalances in wireless sensor networks (WSNs). It effectively outlines the use of Prim’s algorithm for constructing minimum spanning trees (MSTs) to address this issue. The introduction and literature review are well-developed, establishing a solid foundation for the novelty of the approach. However, the article could benefit from a more detailed description of the network initialization and energy consumption modeling to enhance understanding of the setup and baseline assumptions. The following research works can be cited during the revision to enhance the overall quality of this research work:

1. N. Kumar, P. Rani, V. Kumar, P. K. Verma, and D. Koundal, “TEEECH: Three-Tier Extended Energy Efficient Clustering Hierarchy Protocol for Heterogeneous Wireless Sensor Network,” Expert Systems with Applications, vol. 216, p. 119448, Apr. 2023, doi: 10.1016/j.eswa.2022.119448.
2. R. Priyadarshi, “Energy-Efficient Routing in Wireless Sensor Networks: A Meta-heuristic and Artificial Intelligence-based Approach: A Comprehensive Review,” Archives of Computational Methods in Engineering, vol. 31, no. 4, pp. 2109–2137, Jan. 2024, doi: 10.1007/s11831-023-10039-6.
3. I. Surenther, K. P. Sridhar, and M. Kingston Roberts, “Maximizing energy efficiency in wireless sensor networks for data transmission: A Deep Learning-Based Grouping Model approach,” Alexandria Engineering Journal, vol. 83, pp. 53–65, Nov. 2023, doi: 10.1016/j.aej.2023.10.016.
4. Del-Valle-Soto, C., Rodríguez, A. and Ascencio-Piña, C.R., 2023. A survey of energy-efficient clustering routing protocols for wireless sensor networks based on metaheuristic approaches. Artificial Intelligence Review, 56(9), pp.9699-9770.
5. M. K. Roberts and J. Thangavel, “An optimized ticket manager based energy‐aware multipath routing protocol design for IoT based wireless sensor networks,” Concurrency and Computation: Practice and Experience, vol. 34, no. 28, Oct. 2022, doi: 10.1002/cpe.7398.

Experimental design

The experimental design described in the paper seems robust, with the methodology encompassing network initialization, energy modeling, MST construction, and mobile sink optimization. It is commendable that the authors have used diverse datasets for testing, which strengthens the validity of the results. However, the specifics of these datasets—such as their characteristics and why they are appropriate for this study—could be elaborated upon to better assess the applicability of the results across different WSN scenarios.

Validity of the findings

The findings reported indicate that the proposed Prim’s algorithm-based approach effectively achieves energy balance and extends the network's lifespan. The use of key metrics such as Energy Consumption Parity (ECP), Energy Efficiency (EE), Network Lifetime, Throughput, and Packet Delivery Ratio (PDR) for evaluation is appropriate and comprehensive. Nonetheless, the paper would benefit from a comparison with other existing energy balancing techniques to contextually position the effectiveness of the proposed method. A statistical analysis of the performance metrics could also be included to provide a stronger basis for the claimed improvements.

Additional comments

The paper should discuss the scalability of the proposed solution and its feasibility in real-world deployments. Insights into computational overheads, real-time applicability, and compatibility with existing network protocols would be valuable.

While the paper briefly mentions future work, a more detailed discussion on the limitations of the current study would be useful. This might include potential challenges in implementing the algorithm in highly dynamic environments or under varying environmental conditions.

Additional graphical representations and visualizations of the network operations, energy distributions, and before-and-after scenarios with the application of the MST could enhance the comprehensibility and impact of the results.

Information about the simulation tools and software used, along with the settings for Prim’s algorithm implementation, would aid in replicability and validation of the study by other researchers.

Overall, the research presents a promising approach to addressing energy imbalances in wireless sensor networks, with significant implications for enhancing network efficiency and sustainability. Further detailing in certain aspects of the study could make the article even more robust and impactful.

---

## Round 0.2 · Minor Revisions

Please address the final minor point raised by the reviewer.

Reviewer 4 ·

Basic reporting

Kindly include the statistical calculation (Anova calculation) briefly during the revision. I have nothing to add further. This work can be accepted.

Experimental design

Revisions are carried out as per the queries.

Validity of the findings

Kindly do the statistical reporting to solidify your research work.

Additional comments

Nil

---

## Round 0.3 · accepted · Accept

Congratulations. The reviewers are satisfied with the current version of the manuscript.

Reviewer 4 ·

Basic reporting

The required corrections have been carried out by the authors. I don't have any thing to add further.

Experimental design

The required corrections have been carried out by the authors. I don't have any thing to add further.

Validity of the findings

The required corrections have been carried out by the authors. I don't have any thing to add further.

Additional comments

The required corrections have been carried out by the authors. I don't have any thing to add further.